# “It’s Been Ugly”: A Large-Scale Qualitative Study into the Difficulties Frontline Doctors Faced across Two Waves of the COVID-19 Pandemic

**DOI:** 10.3390/ijerph182413067

**Published:** 2021-12-10

**Authors:** Sophie Harris, Elizabeth Jenkinson, Edward Carlton, Tom Roberts, Jo Daniels

**Affiliations:** 1Department of Psychology, University of Bath, Bath BA2 7AY, UK; slh92@bath.ac.uk; 2Health and Social Sciences, University of the West of England, Bristol BS16 1QY, UK; elizabeth2.jenkinson@uwe.ac.uk; 3North Bristol NHS Trust, Bristol BS10 5NB, UK; Ed.Carlton@nbt.nhs.uk (E.C.); tomkieranroberts@gmail.com (T.R.); 4Trainee Emergency Research Network (TERN), Royal College of Emergency Medicine, London EC4A 1DT, UK; 5Bristol Medical School, University of Bristol, Bristol BS8 1UD, UK

**Keywords:** COVID-19, frontline workers, healthcare workers, qualitative research, moral injury

## Abstract

This study aimed to gain an uncensored insight into the most difficult aspects of working as a frontline doctor across successive COVID-19 pandemic waves. Data collected by the parent study (CERA) was analysed using conventional content analysis. Participants comprised frontline doctors who worked in emergency, anaesthetic, and intensive care medicine in the UK and Ireland during the COVID-19 pandemic (*n* = 1379). All seniority levels were represented, 42.8% of the sample were male, and 69.2% were white. Four themes were identified with nine respective categories (in parentheses): (1) I’m not a COVID hero, I’m COVID cannon fodder (exposed and unprotected, “a kick in the teeth”); (2) the relentlessness and pervasiveness of COVID (“no respite”, “shifting sands”); (3) the ugly truths of the frontline (“inhumane” care, complex team dynamics); (4) an overwhelmed system exacerbated by COVID (overstretched and under-resourced, constant changes and uncertainty, the added hinderance of infection control measures). Findings reflect the multifaceted challenges faced after successive pandemic waves; basic wellbeing needs continue to be neglected and the emotional impact is further pronounced. Steps are necessary to mitigate the repeated trauma exposure of frontline doctors as COVID-19 becomes endemic and health services attempt to recover with inevitable long-term sequelae.

## 1. Introduction

The 2019 Novel Coronavirus (COVID-19) pandemic has caused global devastation with over 4.9 million deaths reported to the World Health Organisation (WHO) at the time of writing (October 2021) [1]. The critical role of frontline doctors and healthcare workers (HCW) more broadly during the pandemic cannot be understated. However, this has not come without cost; it has been predicted that at least 115,000 of the recorded deaths due to COVID-19 have been in HCW [2]. In addition to infection risks [3,4], substantial evidence has illustrated the psychological impact of working on the COVID-19 frontline, with high rates of psychological distress and traumatic stress being found in HCW globally [5,6,7,8,9,10]. These findings mirror morbidities observed in frontline staff during previous infectious disease outbreaks [11], which reflect high risk of long-term psychological sequelae without timely intervention [12,13]. 

Various guidelines have been issued during the pandemic with recommendations on how to protect HCW wellbeing [14,15,16,17,18]. However, many of these were developed rapidly when little was understood about the experiences of those working on the COVID-19 frontline [19]. Research has since shown that there is a misalignment between what frontline staff perceived as being important and the recommendations that were prioritised in these initial wellbeing guidelines [18], emphasising the importance of attending to the lived experiences of HCW during the pandemic in order to better understand how to mitigate the inevitable impact of working on the frontline [18,20].

Qualitative evidence reporting on HCW experiences during the first wave of the COVID-19 pandemic highlighted the occupational and psychological pressures associated with working on the frontline [20]. Common themes included high workloads; fear of infection to self, family and loved ones; inadequate personal protective equipment (PPE); lack of training to cope with changing occupational demands; and moral injury (the distress experienced in response to clashes with moral codes [21]) [20,22,23,24,25,26,27,28,29,30]. These themes transcend the COVID-19 pandemic, echoing those drawn out from HCW experiences during previous infectious disease outbreaks such as Ebola and SARS [20]. Similar challenges have also been recorded in the quantitative literature and have consistently been shown to be associated with poorer mental health outcomes for HCW including post-traumatic stress and psychological distress during the COVID-19 pandemic [9,10,31].

Qualitative research to date has primarily explored HCW experiences in the COVID-19 pandemic using traditional semi-structured interviews, but there is evidence to suggest that important insights are being missed, potentially due to participants self-censoring their accounts [23]. Stigma [32], loyalties, and fear of legal/organisational repercussions could result in HCW concealing the less socially desirable aspects of the pandemic during interviews [23]. Gaining insight into these experiences, no matter how ‘ugly’, is crucial in order to learn from the pandemic and mitigate future risks.

Bennett et al. [23] were able to gain “uncensored access to their stories” (p. 6) by encouraging HCW to anonymously audio-record their experiences in the Covid-19 pandemic using an online platform, enabling the researchers to discover new themes not previously identified in the literature. ‘Positive phenomena’ of the pandemic, such as increased social support and post-traumatic growth [24,25,26] were absent from the accounts recorded by Bennett et al. [23], indicating that when unprompted by a researcher, HCW may focus primarily on the negative aspects of working during the pandemic. This highlights the benefits of an added layer of anonymity when collecting sensitive qualitative data, as limiting researcher interaction is proposed to reduce the risk of social desirability bias [33], and emphasizes the need to attend to the experiences which matter most to HCW, the challenges of the frontline. However, the findings have limited transferability and resonance as their sample size was small (*n* = 54) and participants were recruited through social media [34]. Further research which captures a larger more representative sample is needed. 

Another limitation of the current qualitative evidence base is the paucity of research exploring HCW experiences specifically during the second wave of the pandemic [30]. At the time of writing (October 2021) the UK experienced successive pandemic waves with the disease now becoming endemic, the first in Spring 2020 and the second in Winter 2020, with the deadliest day and the highest number of hospital admissions being observed during the second wave [35,36]. Although the National Health Service (NHS) has been strained over many years [37,38], the pressures experienced in the second wave were unparalleled, with three quarters of doctors reporting that the second wave had been busier than the first [39], making it uniquely significant as a period of study. This raises significant concerns for wellbeing as the third wave approaches. 

Looking to evidence from quantitative research, findings suggest that the second wave had clear psychological repercussions for frontline doctors in the UK and Ireland. From the first to the second wave the prevalence of psychological distress for this group increased from 44.7% to 53.2% and psychological trauma from 22.7% to 28.4% [7,8,9]. Without qualitative inquiry it is difficult to understand the meaning behind these findings. Further research is needed to gain a deeper understanding of the experiences of frontline doctors across both the first and the second wave of the COVID-19 pandemic and, more specifically, accounts of the challenges they faced, unprompted and in their own words. 

### Study Aims 

This study aimed to gain an uncensored insight into the most difficult aspects of working as a frontline doctor in the UK and Ireland across both the first and second wave of the COVID-19 pandemic.

## 2. Materials and Methods

This is a qualitative sub-study of the COVID-19 Emergency Response Assessment (CERA) study [7,8,9,40], delivered by the Trainee Emergency Research Network. CERA is an ongoing longitudinal study investigating the presentation and prevalence of distress in frontline doctors during the COVID-19 pandemic in the UK and Ireland. Data for CERA has been collected using online Research Electronic Data Capture (REDCap) surveys which have been distributed to participants during acceleration and deceleration phases of the pandemic. The present study reports on qualitative data gathered during the second wave of the pandemic as part of the fourth CERA survey distributed to participants.

### 2.1. Measures

From the fourth CERA survey [8], all data from the General Health Questionnaire (GHQ-12) [41] and the Impact of Events Scale-Revised (IES-R) [42] and demographic questionnaire were extracted and used to describe the sample. The qualitative data used for primary analysis in the study was taken from a single open-ended question, detailed below. 

#### 2.1.1. Qualitative Measure

The qualitative data used in this study was derived entirely from a single open-ended question, which asked: ‘Please tell us what aspects of working in the pandemic you found particularly difficult?’. This question was designed to elicit reflections on challenges experienced across the pandemic and was not limited by character or accompanied by any prompts. The question was positioned towards the end of the survey.

#### 2.1.2. Quantitative Measures

Quantitative data collected during the fourth CERA survey has been reported in full elsewhere [8]; however, demographic material of those who answered the single-item question stated above was collated for those who participated in this sub-study. Demographic information included participants’ gender, age range, ethnicity, parent speciality, and seniority level.

The GHQ-12, a 12 item self-report measure developed to screen for psychological morbidity [41], has demonstrated high internal reliability and validity across a range of populations [43,44].

The IES-R is a 22 item self-report measure which is used to screen for traumatic stress [42]. The IES-R has been found to have high internal consistency and construct validity [45] and has been widely used during this and other pandemics to screen probable post-traumatic stress symptoms in HCW [10,46].

### 2.2. Participants

The CERA study recruited medical doctors working in emergency medicine (EM), in the intensive care unit (ICU) and in anaesthetics (AN) during the sampling period (first wave of COVID-19 pandemic) in the UK and Ireland; non-doctors and those not working in EM, ICU or AN during the sampling period were excluded [7,8,9,40]. Full details of the initial recruitment procedure can be found in the CERA study protocol [40]. 

To be included in the present study, participants needed to have completed the fourth CERA survey [8], provided a text response to the qualitative question and have indicated consent to both of the following statements ‘I agree for the CERA data to be shared with other ethically approved research projects (yes/no)’ and ‘I agree for anonymised data to be shared with other researchers (yes/no)’. Those who did not consent to both of these statements were excluded from the present study. 

### 2.3. Procedure

The fourth CERA survey opened in the UK on 28 January 2021 and closed on 11 February 2021, and in Ireland it opened on 1 February 2021 and closed on 15 February 2021 [8]. Data from participants who indicated consent to both statements were collated and anonymised by the CERA principle investigator (TR) before transferring to the principle investigator of this study (SH) for analysis. All data were stored in accordance with the University of Bath Data Security and Confidentiality Policy and the Data Protection Act 2018.

### 2.4. Planned Analysis

This study followed an interpretivist paradigm to facilitate an inductive sensemaking process, adopting the perspective that the nature of reality is socially constructed [47]. Analysis was guided by Hsieh and Shannon’s [48] conventional content analysis approach to allow categories to flow directly from the data. Content analysis was chosen as it permits the analysis of large amounts of data [49] and has been widely used to understand HCW experiences during the pandemic [28,29,30]. 

Analysis was conducted by SH, with input from EJ an experienced qualitative researcher in the field and health psychologist. First, SH engaged in multiple readings of the data for familiarisation and initial impressions were noted. Next, SH coded the first 100 extracts to develop a coding scheme; this was checked by EJ to ensure fit to the data. This scheme was then applied to code the entire dataset using NVivo 12 Pro (QSR International Pty Ltd., Doncaster, Australia) with new codes added if data did not fit within the existing scheme. EJ then double coded 100 extracts to increase robustness of the analysis and any divergent opinions were reviewed and codes revised. Finally, codes were categorised, and these categories were latently analysed to develop themes. 

SH kept a reflexive diary throughout analysis to help improve trustworthiness of interpretation [50]. SH has had no contact with participants and does not know any frontline doctors personally. However, SH has had experience working on a similar research project and was mindful that prior familiarity can influence interpretation of the data [51]; SH ensured reflection on this during analysis.

### 2.5. Ethical Approval

CERA was sponsored by North Bristol NHS trust and received ethical approval from the University of Bath (reference: 4421) and the Ethics Committee at Children’s Health Ireland at Crumlin and received regulatory approval from the Health Regulation Authority and Health and Care Research Wales (IRAS: 281944). The present study was granted ethical approval by the University of Bath Psychology Research Ethics Committee (references: 21–138) and was sponsored by the University of Bath and North Bristol NHS trust.

## 3. Results

Of the 1719 participants who responded to the fourth CERA survey [8], 1384 provided consent for their data to be shared with this study (80.5%). Of those, four did not provide a text response and one indicated that the open-ended question was not applicable. A total sample of 1379 participants remained (80.2% of the original sample) all of which were included in analysis.

### 3.1. Sample Characteristics 

Demographic and psychometric data are reported in Table 1. All seniority levels were represented, with 42.8% of the participants male and 69.2% were white. Nearly a third of participants (32%) had an IES-R score indicating the presence of post-traumatic stress symptoms (≥24). To assess pattern of missing data in the IES-R and GHQ-12, Little’s test of Missing Completely at Random (MCAR) [52] test was performed and was found to be non-significant for items in the IES-R χ^2^ = 719.7, DF = 858, *p* = 1.000 and the GHQ-12 χ^2^ = 179.2, DF = 221, *p* = 0.982, indicating that the data were MCAR. Due diligence manual calculation and imputation of the median score did not alter the descriptive statistics for the total questionnaire scores. Listwise deletion was therefore used during analysis. 

### 3.2. Analysis of Qualitative Data

Responses to the single open-ended question ranged from 1 to 575 words, with a median of 21 words per response (IQR = 10,37). Four main themes were identified: I’m not a COVID hero, I’m COVID cannon fodder; the relentlessness and pervasiveness of COVID; the ugly truths of the frontline; and an overwhelmed system exacerbated by COVID. Themes, categories and example quotes can be seen in Table 2. Participants have been identified by gender and professional grade, and when differing viewpoints have been identified in text the corresponding quote numbers have been provided.

#### 3.2.1. I’m Not a COVID Hero, I’m COVID Cannon Fodder

This theme relates to frontline doctors feeling as though their wellbeing had been disregarded during the pandemic and encompasses two categories: exposed and unprotected; and “a kick in the teeth”. The first speaks more to doctors’ perceptions of safety on the frontline, whereas the second encompasses doctors’ reflections on the actions of those external to the frontline. 

##### Exposed and Unprotected 

Many participants reported feeling unsafe and inadequately protected on the frontline, with fears of infection and transmission being commonly reported. Accounts of staff becoming infected, seriously ill and in the worst cases dying illustrate the palpable threat to safety. Perceived risks included inadequate PPE; staff and patient none-compliance with hospital safety measures; and delayed vaccinations. Those who spoke of the vaccine rollout conveyed the unfairness of how it was handled, with non-frontline staff appearing to be prioritised, and second vaccinations cancelled at short notice. This left a minority questioning the integrity behind the reason for the vaccine delays. These actions as well as inactions, resulted in anger, anxiety, and the feeling that frontline staffs’ safety had been overlooked.

##### “A Kick in the Teeth”

Participants felt as though the actions and attitude of the Government, NHS trusts and the public were not in support of frontline workers and did not reflect the gravity of the situation. Reports included feeling as though the Government had not acted enough nor acted in the best interests of frontline staff, with frustrations around poor leadership decisions, not enforcing tighter restrictions, PPE procurement, and delaying second vaccinations. 

Similar criticisms were raised regarding the lack of support and poor decisions made by NHS trusts, with additional concerns relating to the lack of clear communication from “invisible” management teams. Of particular concern to junior doctors was the disruption to their training; exams were cancelled, training opportunities depleted, and pressures to complete training requirements continued, in the face of what felt like little understanding and support. 

Lastly, some participants expressed anger and hurt that people continued to break lockdown rules, noting a change in general attitudes towards the pandemic; particularly distressing was those who deny the pandemic’s existence. Overall, there was a real sense of alienation from non-HCW, with frontline staff feeling disregarded, betrayed, and left to fight the COVID-19 pandemic alone. 

#### 3.2.2. The Pervasiveness and Relentlessness of COVID

At the time of the fourth CERA survey the pandemic had been on-going for just under a year, with many participants working across both the first and the second wave. This theme captures participants reflections on the enduring nature and inescapability of the pandemic, comprising of two categories: “no respite”; and “shifting sands”.

##### “No Respite”

Numerous participants described their workload and the pandemic more generally as “relentless” and “never ending”. Accounts indicate that over duration of the pandemic there were limited opportunities to decompress outside of work due to numerous factors including cancellation of annual leave, restrictions to recreational activities, and external pressures such as home schooling. This left many “in the unsustainable position of emotional loading with no outlet” (#119, M, junior doctor) with reports of burnout symptoms, exhaustion and general psychological distress being common. Especially impactful was the loss of social interaction with friends, family, and work colleagues, leading to some doctors’ feeling lonely and isolated. Added to these pressures was the reality that COVID was everywhere, at work, at home, in the media—there was “no respite” and “no escape”. 

In contrast, a small minority of participants reported no difficulties during the pandemic with a few describing positive experiences, indicating that although the majority found the pandemic relentless and challenging, others did not (see Table 2, quote 2.vi).

##### “Shifting Sands” 

Some participants reflected on their experiences across the different waves of the pandemic. Within these reflections were comments indicating that the first wave felt more uncertain and the second more relentless, with one person stating “Last year, the unknown and unceratainty (uncertainty). This year the never ending” (#139, F, senior doctor). Some noted a change in roles across the pandemic, often resulting in increased or reduced feelings of usefulness. Others compared difficulty levels across the waves, the majority of whom reported the second wave as being more difficult. Reasons included increased deaths, younger patients, the relentlessness, and feeling less supported. Nevertheless, a small proportion did report seeing improvements compared to the first wave such as less uncertainty, improved processes, and increased team cohesion.

#### 3.2.3. The Ugly Truths of the Frontline

This theme embodies the ‘ugliness’ of working on the COVID-19 frontline, capturing the emotive, distressing and often unseen challenges doctors faced. This theme contains two categories: “inhumane” care and complex team dynamics. 

##### “Inhumane” Care

Many participants discussed the unpleasantness of providing patient care during the pandemic, with challenges including complex decision making, increasingly younger patients, and the acuity of illness. Care for COVID patients was repeatedly depicted as being futile due to limited treatment options and the difficulties with delivering a “good death”. Accounts were often candid, detailed, and emotive, leaving a sense that participants wanted the reader to truly ‘see’ the realities of working on the frontline. This included care being described as “torture”, “brutal”, and “inhumane”, indicating the torment some doctors felt about the patient experience during the pandemic. 

An important factor related to this was the visitation restrictions, meaning families were not able to be involved in patient care in the way they would normally expect to be. Some participants comments on this were brief and related to communication challenges, whereas other participants’ reflected on the distressing nature of breaking bad news down the telephone as well as watching patients suffer, and in the worst cases, die alone (see Table 2, quotes 3.i, iii, iv). Feelings of guilt and sadness were common, with some participants indicating that they had been traumatised by their experiences caring for patients.

However, it was not just the patients who experienced “inhumane” care on the frontline, as a small minority of participants disclosed experiencing mistrust, aggression, and abuse from patients and relatives. Furthermore, several participants reported problems with patients and relatives not complying with infection control measures in hospitals, placing staff at unnecessary risk.

##### Complex Team Dynamics 

A common depiction within accounts was the sense that participants felt both literally and/or figuratively distanced from their colleagues during the pandemic. Factors related to this included the pressure of working in an emotionally charged environment as well as the separation of colleagues due to social distancing, shielding, and redeployments. Of those who spoke of their colleagues, the majority expressed concerns for their physical and emotional wellbeing, and with this often came a sense of responsibility as well as powerlessness to help. It was clear from some accounts that it was incredibly upsetting to see their colleagues struggling. 

On the other hand, others expressed fractious relationships, with repeated reports of lower team morale and colleagues being snappier with one another. Frustrations ranged from minor to more serious, with some reporting feeling unsupported by colleagues’ actions such as none-compliance with infection control measures, and others reporting instances of “bullying” and “aggression”. A common perception expressed was that some of the team had not “pulled their weight”, resulting in frustration for those who felt like they were contributing more to the pandemic efforts, and expressions of guilt and uselessness for those who felt as though they had not done enough. From these accounts, there was a sense that for some only those who were working directly on the frontline (i.e., treating COVID patients in ICU) were considered the true ‘heroes’ of the pandemic. 

#### 3.2.4. An Overwhelmed System Exacerbated by COVID 

This theme represents organisational challenges frontline doctors faced with regards to their working environment during the COVID-19 pandemic. This includes pre-existing problems in the NHS as well as the addition of new challenges related to the pandemic. This theme consists of three categories: overstretched and under-resourced; constant changes and uncertainty; and the added hindrance of infection control measures. 

##### Overstretched and Under-Resourced

Many participants reported problems with understaffing and high workload. Factors related to this included increased volume of high acuity patients and the loss of staff to redeployment, sickness and shielding. This was reported as placing unprecedented demands on those left working on the frontline including working long hours and picking up additional shifts. Difficulties with capacity and physical resources were also frequently reported and predominantly pertained to the ED. Participants spoke of lack of flow and overcrowding in ED resulting in corridor medicine and some needing to treat patients in ambulances. Accounts detailed non-COVID patients who presented to services either as acutely unwell due to delaying seeking medical treatment or with ailments that would be better treated in the community. With not enough space and resources for everyone, concerns regarding the standard of care being provided and growing waiting lists were voiced. 

##### Constant Changes and Uncertainty

Participants described being required to work flexibly, with “constantly changing” guidelines, rotas, and roles. Accounts indicate that these changes were happening frequently, rapidly, and often without clear communication or consent. Descriptions of feeling uncertain were common, and it was clear that for some the changes made them feel on edge and out of control.

##### The Added Hindrance of Infection Control Measures

Although necessary, infection control measures seemed to make an already difficult job even harder. Many participants reported challenges with wearing PPE including inconvenience, severe discomfort and difficulties communicating. Less cited, but seemingly just as disruptive, were the social distancing measures at work, making handovers and debriefs more difficult as not all team members were allowed to be in the room at once. Accounts indicate that participants were not able to perform to the best of their abilities due to these constraints. 

## 4. Discussion

The aim of this study was to gain an uncensored insight into the most difficult aspects of working as a frontline doctor in the UK and Ireland across the first and second wave of the COVID-19 pandemic. Qualitative data from a large sample of frontline doctors was analysed and four key themes were identified. Themes encompassed participants’ concerns that frontline staff safety and wellbeing had been repeatedly overlooked; the relentlessness of the pandemic; the distressing and often ‘ugly’ nature of patient care and teamwork; and the organisational challenges which often impeded frontline doctors’ work performance. These findings offer a comprehensive and highly emotive account of the most difficult aspects of working as a frontline doctor during the COVID-19 pandemic that has not yet been reported to this extent. Findings communicate a sense that, for many, the relentlessness of a second wave, without reprieve, was more challenging physically and emotionally, representing worrying findings given the current context of an approaching third wave. 

Findings from this study echo themes drawn out in earlier, first wave qualitative research [22,23,24,25,26,27,28,29,30], providing evidence of the persistence of these problems into the second wave of the pandemic, indicating that little has been done to address serious concerns about working practices raised from the first wave [20,22,24]. Yet evidence from these uncensored accounts highlight that these pressures had only intensified during the second wave, owing in part due to the length of time participants had been exposed to them and the lack of time to rest and recuperate. Previous research has shown that increased time spent working on the COVID-19 frontline is associated with higher levels of stress [54], and this resonated with accounts from doctors in this study. Reflections on the “relentlessness” of the pandemic were common, and this represented a primary stressor for participants in the second wave, with many voicing a clear and desperate need for respite. 

Another key source of stress for participants was the fear of becoming infected with the virus. This has been a constant theme in HCW experiences throughout the research [20,22,23,24,26,27,29,30], transcending different countries, different pandemics [20], and now different pandemic waves. Consistent with research conducted during the first wave [22,24,30], participants reported not having access to adequate PPE during the second wave, highlighting the continuation of this problem across the pandemic, which will have exacerbated raised concerns about personal safety and transmission to families, key predictors of mental health in a recent longitudinal study [9]. This finding is also concerning given evidence that appropriate use of PPE offers adequate protection from infection [55], raising the difficult question as to whether enough was done to protect the many frontline staff who lost their lives during first and then further in the second wave, having already protested at life-saving PPE shortages [7,56,57]. 

Participants also expressed discontent and perceived betrayal at the increased exposure to risk during the second wave as the UK Government extended the gap between vaccination doses from three to 12 weeks [58]. This meant that many doctors faced delays to their second vaccination [39], despite evidence at the time indicating that the immune response was weaker following only one vaccine dose compared to two [59]. Due to the paucity of qualitative research reporting on HCW experiences in the second wave, reflections on the vaccination delays are not represented in previous research and add a unique contribution to the literature; participants’ accounts conveyed the fear and anger some felt in response to this decision, with a sense that the vaccination delays as well as other perceived risks, such as PPE provision, exemplified that the UK Government placed little to no importance on frontline staff safety. 

Similar sentiments regarding the UK Government’s handling of the pandemic have been found elsewhere in the research, with studies describing feelings of anger and feeling let down by those in authority [19,23,60]. A recent qualitative study conducted by French et al. [60] equated these feelings to moral injury, adopting Shay’s [61] definition which is characterised as a betrayal of perceived morality by a person in authority. This definition resonates here, with many participants describing feeling unsupported and disregarded by the Government, NHS trusts and non-clinical management teams. French et al. [60] state that “if moral repair is to take place across the public sector, it will be vital for those leading the country to acknowledge and atone for their mistakes” (p. 5), arguing that without moral repair, other strategies to support HCW recover from the pandemic may be less effective. The incidence of betrayal-based moral injury found in the present study indicates that this phenomenon warrants further consideration when designing post-pandemic recovery strategies; furthermore, the finding that betrayal experienced may vary by seniority level, such as the impact junior doctors felt the pandemic had on their training, suggests this may need to be tailored by professional grade. 

Accounts in the present study also point to instances of perpetration-based moral injury, which is characterised by feelings of guilt associated with actions or inactions which violate an individual’s moral code [21]. This can be seen in participants’ descriptions relating to patient care. Higher reported exposure to moral injury has been found to be strongly associated with increased levels of anxiety, depression, post-traumatic stress symptoms and alcohol misuse [31]; however, to date no validated treatment for moral injury exists [62], indicating a clinical need which urgently needs addressing. An array of psychological models designed to target moral injury have been proposed [62,63], but further research trials are needed to explore the efficacy of these interventions to devise an evidence-based model of care.

Another concerning finding from the present study relates to frontline doctors’ perceptions of peer and public support. The qualitative literature on social support during the COVID-19 pandemic has been mixed, with some research suggesting that HCW felt in receipt of more support from their colleagues and wider society during the pandemic [24,25,26], and other studies noting a more complex relationship between HCW and social support [23,64]. Those who participated in this study align more closely with the latter, as accounts regarding social support were overwhelmingly negative. This may reflect anonymous uncensored responses without concerns for potential consequences. Social support has been shown to be a protective factor for adverse mental health outcomes in HCW during the pandemic [10,65], with one third of junior and senior doctors reporting it as a key coping strategy [66], highlighting the need for implementation of formal and informal peer interventions for all professional grades to ensure that frontline doctors feel supported going forwards. 

The COVID-19 Clinician Cohort study (CoCCo) [19] developed empirically grounded recommendations and a model of psychological care derived from the accounts of psychologically distressed frontline doctors, purposefully sampled to represent a range of personal and professional characteristics. This model encompasses and addresses concerns raised by and echoed here, with emphasis placed on meeting basic needs such as ensuring access to adequate PPE and allowing doctors time to decompress, as well as facilitating access to peer support and specialist interventions. This stepped pathway of care provides the most coherent model to date that can be implemented into services to better support frontline doctors into the future; however. policy makers and clinical managers need first to recognize the absolute necessity of intervention.

The public health implications of the findings from the present study cannot be over-emphasised. Many of the challenges reported by the frontline doctors here have been shown to be associated with higher rates of psychological morbidities in HCW during the COVID-19 pandemic [9,10,13] and research has found that doctors with poorer mental health are more likely to report providing suboptimal patient care [67] and making major medical errors [68], highlighting the importance of nurturing a psychologically well healthcare workforce. Moreover, factors such as high workloads, the Government’s handling of the pandemic, and inadequate PPE have been commonly cited as reasons that frontline doctors as well as HCW more broadly are considering leaving the profession [69,70,71]. As waiting lists continue to grow and a third wave approaches, preventing a staff exodus is vital. It is therefore crucial that frontline doctors’ voices are not only heard but responded to, representing a further call to action, a repetition of many such earlier calls, to ensure the physical and psychological safety of frontline doctors.

### Strengths and Limitations

This study reports on one of the largest qualitative datasets relating to frontline workers experiences in the COVID-19 and other previous pandemics. Similar to the study conducted by Bennett et al. [23] which claimed to gain “uncensored access” (p. 6) to HCW stories, participants did not meet with researchers, and instead provided qualitative responses using an online platform. This allowed for a breadth of raw and unprompted responses, which ensured findings represented the difficulties which mattered most to frontline doctors. Findings amplify the concerns raised in previous research and add considerable value to the literature by highlighting the persistence of these problems into the second wave. Moreover, the sample represented a diverse range of personal and professional characteristics, including individuals commonly under-represented in qualitative research such as men [72] (42.8%) and those from ethnic minority backgrounds [73] (excluding white minorities; 17.0%), increasing confidence in the findings reported here as well as their relevance to these groups. 

However, as 52% of NHS doctors are male [74], their views are still not adequately represented in the present study. Considering that psychological risk has been shown to vary by gender [6,10,31], it seems pertinent that greater effort is taken to engage male HCW in future qualitative research to better understand the full breadth of experiences during the pandemic. Furthermore, as this study focused solely on the difficult experiences of frontline doctors, findings may not represent the views of HCW more broadly. Evidence has shown that mental health outcomes in the pandemic vary by professional group [5,6,31], meaning further research is needed to gain insight into the experiences of other HCW groups following two waves of the pandemic. 

## 5. Conclusions

Frontline doctors faced a multitude of challenges across the COVID-19 pandemic, many of which had been identified as being problematic during the first wave [22,23,24,25,26,27,28,29,30] and continued to persist into the second despite repeated calls to action. The ‘ugly’ and uncensored truth reflects these, and possibly many other frontline doctors feel angry, betrayed and unsupported—through vaccination delays, inadequate PPE and working through the strain on a system already overburdened. 

These problems urgently need addressing as COVID-19 becomes endemic and health services attempt recovery, where the repeated exposure to these challenges and absence of reprieve are likely to bear long term consequences. Action is needed to ensure that frontline doctors feel supported, moral injuries are repaired, and further risks to safety and wellbeing are mitigated.

## Figures and Tables

**Table 1 ijerph-18-13067-t001:** Demographic and psychometric data.

Demographic Information	*n* = 1379 (%)
Age	
20–25	32 (2.3)
26–30	282 (20.4)
31–35	286 (20.7)
36–40	218 (15.8)
41–45	189 (13.7)
46–50	144 (10.4)
51–55	124 (9.0)
56–60	74 (5.4)
61–65	25 (1.8)
66–70	5 (0.4)
Gender	
Male	590 (42.8)
Female	742 (53.8)
Other	5 (0.4)
Missing	42 (3.1)
Ethnicity	
White	954 (69.2)
Mixed or Multiple ethnic groups	35 (2.5)
Asian or Asian British	160 (11.6)
Black, African, Caribbean or Black British	25 (1.8)
Other ethnic group	15 (1.1)
Missing	190 (13.8)
Seniority	
Junior doctor	390 (28.3)
Middle grade doctor	261 (18.9)
Senior doctor (consultant grade)	560 (40.6)
Other senior doctor	104 (7.5)
Other doctor grade	64 (4.6)
Parent Speciality	
Emergency medicine	570 (41.3)
Anaesthetics	535 (38.8)
Intensive care medicine	137 (9.9)
Other	185 (13.4)
Psychometric Measures	
IES-R	
Median (Q1,Q3)	16 (7.30)
Range	0–88
PTSD is of clinical concern ≥ 24 *n* (%)	441 (32.0)
Probable PTSD ≥ 33 *n* (%)	275 (19.9)
Missing *n* (%)	98 (7.1)
GHQ-12 (0-1-2-3)	
Median (Q1,Q3)	16 (12.20)
Range	1–36
Missing *n* (%)	42 (3.0)

Note: PTSD, Post-traumatic stress disorder. Junior doctors: F1, foundation year 1; F2, foundation year 2; ST1–3, general practitioner trainee/specialist trainee years 1–3, F2-ST3, clinical fellow. Middle grade doctor: ST4–8, specialist trainee/clinical fellows years 4–8. Senior doctor: consultant/associate specialist/staff grade/general practitioner/certificate of eligibility for specialist registration.

**Table 2 ijerph-18-13067-t002:** Themes, categories, and example quotes.

Theme	Categories	Example Quotes
I’m not a COVID hero, I’m COVID cannon fodder	Exposed and unprotected	“Still having PPE below WHO standards i.e., no FFP3 masks for standard use, no protective eye wear—I had to buy my own goggles and using those plastic aprons while the Far Eastern doctors have full body suits to do even swab. Plus no negative pressure zones in my ED.” (#112, M, other senior doctor) “Did not feel good when loads of patients generating aerosol I was seeing and a lot of staff getting infected.” (#113, M, middle grade doctor) “Angry about how vaccine has been handled…Feel I agreed to first dose under false pretences, having gained informed consent for second dose at 3 weeks I don’t understand how they can then move the goalposts (we would surely lose registration if we did similar to patients with any medication) I believe this strategy is dangerous at an individual level for clinicians who are more at risk than if they had 2 doses and at a population level with risk of mutation…I believe it has been done purely to improve numbers for media purposes and I am so angry that having put our lives at risk for a year we are being forced to be less protected than we could be in terms of ppe and vaccine.” (#114, F, senior doctor) “I feel, at times, that I am considered totally expendable and that if I die or become ill not only will it have been preventable with political will, I will simply be an inconvenient statistic. I’m not a COVID hero, I’m COVID cannon fodder.” (#115, F, other senior doctor)
	“A kick in the teeth”	v. “Knowing the government was failing in so many ways to support us—failed test & trace, failed PPE procurement, weak messaging, permitted non-compliance with mask-wearing and distancing, set a poor example (Barnard Castle, etc.). We as healthcare providers were alone and utterly unsupported. Apart from the weekly round of applause that was a pointless gesture and felt like a kick in the teeth.” (#116, M, junior doctor) vi. “Slow decision making from senior leaders invisibility of some of the executive team who should have been leading us, whilst they still blocked decisions we were making.” (#117, F, senior doctor) vii. “In my experience I think the training programmes have had little sympathy or relaxation for how COVID affects training—all the official guidance says there will be extenuating circumstance but when it comes to progression only the most minor of issues are allowed to be attributed to COVID.” (#118, F, Other doctor grade) viii. “The poor and frankly disrespectful way NHS Trusts have treated junior doctors (cancellation of leave, asking to work “voluntary” shifts, cancelling vaccine appointments for 2nd dose) has me feeling undervalued, disrespected and constantly angry.” (#119, M, junior doctor) ix. “Have felt frustrated when seeing the public blatantly avoiding and not following the rules. It feels a bit disrespectful to ourselves and my colleagues some of whom have sadly lost their lives due to COVID.” (#120, M, senior doctor)
2. The relentlessness and pervasiveness of COVID	“No respite”	“Unrelenting. Groundhog day.” (#132, M, senior doctor) “I am already very tired, worn out, burn out, and this looks like it will never end.” (#133, F, junior doctor) “A major incidence is fine but this has basically been a nearly 12 month major incident. Not one person I have spoken to hasn’t wished for a positive lateral flow test even if their PCR swab is negative just so it would mean a day or two extra off work.” (#134, F, middle grade doctor) “The difficulties of a heavy rota with very little exposure to social activities outside of work (which I personally used as a coping mechanism) has made my risk of burnout increase by a magnitude!” (#135, M, middle grade doctor) “Working with it consistently at work, then when at home it I’m being on news, tv and all anyone can talk about. No escape.” (#136, M, middle grade doctor) “I am working in the vaccine clinic which I find really enjoyable, no unpleasant events or PTSD.” (#137, F, senior doctor)
	“Shifting sands”	vii. “The second/third wave has been much more difficult. Normal presentations have continued at a similar level to normal. Everyone is exhausted and worn out. I’ve found COVID deniers particularly upsetting.” (#138, M, senior doctor) viii. “I was in ED in the first wave and saw a lot of traumatic and distressing scenes…This third lock down I’ve been working (in a different department) have had it relatively easy in comparison to the first wave and to my colleagues. This has left me with feelings of guilt that I’m not doing enough, and working in a different hospital has left me wishing I was where I was before doing the job I did in the first wave so I can help my friends and support them.” (#139, F, junior doctor) ix. “It’s been much better for the 2nd wave. We’ve changed how we manage the anaesthetic workload & we feel more in control of our work. The work is stressful & sad but it is a shared experience & we are talking about it with each other.” (#140, F, senior doctor)
3. The ugly truths of the frontline	“Inhumane” care	“There’s one patient who was only comfortable on 60 litres optiflow but we were running out of oxygen and I insisted he change to CPAP to conserve supplies. He needed intubation and then died and I feel guilty that his last conscious memory was of me torturing him with the CPAP mask. A young mother was admitted to ICU on CPAP and we’d just been given an ipad to help families video call: I kept asking the nurses to help her speak to her family but they delayed until it was too late and we had to intubate her, she died without saying goodby (goodbye).” (#121, F, senior doctor) “People on CPAP getting agitated and needing to physically pin them down and give sedation when you don’t think there is much hope of them getting better.” (#122, M, middle grade doctor) “Communicating bad news to relatives over the phone.” (#123, F, senior doctor) “Telling someone that their loved one is going to die over the phone, and then inviting them in to watch them die, when they have’t (haven’t) seen them for weeks is really traumatic for all.” (#124, F, senior doctor) “I feel guilty all the time now, as I don’t feel like I can be the doctor I would like to be or the doctor I wish would look after my loved ones.” (#124, gender unknown, junior doctor) “The patients are becoming in general increasingly difficult—verbal and physical abuse, spitting, hitting us, threatening us with legal action and a family charged into A&E looking to find me with violent intent obvious. This is not uncommon and becoming increasingly common.” (#125, F, middle grade doctor)
	Complex team dynamics	vii. “Team bonding has been more difficult since we cannot go out together, we have to keep heing (being) aware of the distance, we cannot share food etc.” (#126, M, junior doctor) viii. “My own biggest challenges have been the moral distress of watching colleagues struggle, and worrying about their wellbeing—this has been accentuated by the fact that my own world has been too busy in other related matters to be able to directly offload their workload, leading to feeling inadequate for prolonged spells.” (#127, gender unknown, senior doctor) ix. “Shortage of staff. Decreasing staff morale. Cracks in the team.” (#128, M, Consultant) x. “The consultant body was extremely against supporting the rota, and this has made the department toxic to work in. This behaviour has filtered down to trainees, staff grades and allied staff. It’s been ugly.” (#129, M, middle grade doctor) xi. “Pressure to play a meaningful role—my jobs meant I haven’t encountered many patients with COVID and therefore I feel I am not playing my part.” (#130, F, junior doctor) xii. “The constant noise about how tough the ITU guys have had it has genuinely pissed me off (and I know that is totally unreasonable) because I look at my own specialty (EM) and I think about how bloody awful the last 5 years have been over wintertime—we’ve had patients dying on our corridors and all the trust ever seemed to want to do was apportion blame, so it got hidden and it was frankly fucking soul destroying- so when I’m asked to feel for my colleagues in the ITU I get that I should be sympathetic (and I can see how hard this is for them) but I don’t really feel as though I have anything left…Sorry, I know I’m meant to feel differently and I would if I could. I don’t think I would say this in an open forum though.” (#131, M, senior doctor)
4. An overwhelmed system exacerbated by COVID	Overstretched and under-resourced	“This has been one of the worst winters I’ve ever experienced in my 12 years as a doctor. The bed crisis is shocking and we’ve gone back to the bad old days of patients being on trolleys in A&E for 12 h just waiting for a bed. We waited 8 h for an ITU bed last week, it’s unacceptable.” (#101, F, other senior doctor) “Intensity of long shifts in COVID ICU with very high workload, overstetched [overstretched] staffing. Worst week I palliated 3 patients in one week on call. Felt very sad and a little traumatised.” (#102, M, senior doctor) “Working in hospitals that run near 100% capacity near 100% of the time (prior to the outbreak) and then expecting and trying to take a service that has little slack and stretching it further. It’s been relentless and exhausting, sometimes you are left feeling that despite doing our best we should be doing better but can’t given the circumstances/resources.” (#103, M, junior doctor) “The numbers of unwell patients—many not suffering from COVID 19—who are attending hospital. Many are more unwell than they would have been in 2019 as the out-patient investigations are not happening quickly enough.” (#104, F, senior doctor)
	Constant changes and uncertainty	v. “Ever changing protocols with little to no indication from seniors (consultants or managers) regarding these changes prior or even subsequent to them—nurses definitely seemed to be more in the know than ED registrars.” (#105, F, middle grade doctor) vi. “Frequent changes in work area and pattern. Fear of criticism or litigation when working outside normal practice.” (#106, F, senior doctor) vii. “I have been moved across 3 hospitals within 12 months, requiring me to move home each time. We have been treated like pawns with no thought to how it affects our personal lives.” (#107, M, middle grade doctor)
	The added hinderance of infection control measures	viii. “Wearing PPE, I feel suffocated and experience physical symptoms (headache, overheating) and increased anxiety and brain fog, leading to slow decision making and insecurity and stress.” (#108, F, middle grade doctor) ix. “Trying to communicate with patients when wearing a mask especially the elderly as they can’t hear and unable to lip read. You can’t smile at them to reassure them.” (#109, F, other senior doctor) x. “Angry infection control sisters bursting into handovers to tell us only four, not five people are allowed in a room, compromising safe handovers and making us feel like terrible people.” (#110, gender unknown, junior doctor) xi. “Limited space for breaks and to eat meals due to social distancing measures. Lack of computer space for the same reason” (#111, M, middle grade doctor)

Note: Participants are identified by #, participant number; M/F, gender; professional grade. CPAP stands for continuous positive airway pressure and comprises a mask and hose/or a nose piece to deliver air pressure to patients [53].

## Data Availability

Requests for data will be considered on an individual basis due to the high emotional and personal nature of the content.

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
