# Peer review of "“It’s Been Ugly”: A Large-Scale Qualitative Study into the Difficulties Frontline Doctors Faced across Two Waves of the COVID-19 Pandemic"

_ijerph, 2021, doi:10.3390/ijerph182413067_

Round 1
Reviewer 1 Report
The qualitative study reported in the manuscript aimed at gaining insight into the most difficult aspects of working as a doctor in the UK and Ireland across the first and second wave of the COVID-19 pandemic. Participants were frontline doctors (N=1379) who worked in emergency, anaesthetic, and intensive care medicine. Content analyses revealed four key themes: 1. the perception of (un)safety (“I’m not a COVID hero, I’m a COVID cannon fodder"); 2. the relentlessness and pervasiveness of COVID; 3. the ugly truths of the frontline; 4. an overwhelmed system exacerbated by COVID.
I believe that the topic is extremely important and socially relevant. Furthermore, this study may contribute to sheld light on frontline doctors' experience, which has not been largely investigated during the pandemic. Methodology is accurate and the qualitative approach is appropriately described.
However, there are some points wich might be addressed:
1. how about participants' characteristics? For instance, how about gender? the sample comprises a large number of both men and women and I wonder whether there may be significant gender differences in coping with the pandemic at work.
2. in line with my previous comments, how about the potential influence of professional grade? I think this is a relevant factor as well.
3. which is the relation between qualitative and quantitative data? If I have properly understood, each participant was administered GHQ-12 and IES-R. However, results about these scales are presented only with descriptive statistics. I would suggest to examine more in depth the link between participants' qualitative data and their scores on GHQ and IES.
Reviewer 2 Report
This paper presents a thorough examination of the qualitative perceptions of front-line physicians working during the second wave of COVID-19 in the UK. The rationale for the study was clear, the methods well presented. The results of the one open-ended question yielded fruitful material for a nuanced perspective of working in these unprecedented times. The discussion was clear and relevant. The authors might consider adding in the discussion what kinds of support healthcare workers might benefit from and how to improve safety and well-being of the front-line health care team.
Reviewer 3 Report
Dear Authors,
Congratulations, your article entitled ““It’s been ugly”: A large-scale qualitative study into the difficulties frontline doctors faced across two waves of the COVID-19 pandemic” is interesting and well written. In my opinion, only minor revisions, as for my list here below, are needed before it can be considered for possible publication.
Best regards,
The Reviewer
Abstract
- Line 22: not sure why the sentence “I’m not a COVID hero, I’m a COVID cannon fodder” has been identified as a theme and it is not inside the parenthesis
Introduction
- Lines 58-60: also lack of training in preventing occupational infectious respiratory risk should be added here
Materials and Methods
- Can you confirm to me that the study includes only Medical Doctors?
- Lines 123-4: more details on “question regarding current work location; and a single open-ended qualitative question” are needed.
- Please describe in this section the seniority levels used for the results, as for many readers this information may be not clear and differently interpreted in Countries other than UK and Ireland.
Results
- Table 1: ethnicity: isn’t it better to use Caucasian instead of “white”?
- Table 1: for Psychometric Measures a column with the interpretation of the scores should be added
- Table 2 and paragraph 3.2.1. Same observation as for abstract: I am not sure of the reason why the sentence “I’m not a COVID hero, I’m a COVID cannon fodder” has been identified as a theme, while the other themes are not in quotes and seem representing more general concepts. This sentence is a quote of something said by your sample, and I am not sure that all of them used exactly the same words. I would perform an effort to generalize this sentence (e.g. Perception of not being heroes, but cannon fodder)
Reviewer 4 Report
Congratulations for this paper.
It is an interesting study where authors show the ideas and aspect of working as a frontline doctor in a mix qualitative and quantitative methods, because in many mass media they were called "hero"
It has a good background an excellent methodology.
I understand that this study is part of CERA study protocol. It is very important the ethical Approval and planned Analysis
The sample is 1379 participants which is accurate.
The discussion is perfect and the most important aspect is the qualitative results and these are discussed very well with many references
